# Discovery of New Small Molecule Hits as Hepatitis B Virus Capsid Assembly Modulators: Structure and Pharmacophore-Based Approaches

**DOI:** 10.3390/v13050770

**Published:** 2021-04-27

**Authors:** Sameera Senaweera, Haijuan Du, Huanchun Zhang, Karen A. Kirby, Philip R. Tedbury, Jiashu Xie, Stefan G. Sarafianos, Zhengqiang Wang

**Affiliations:** 1Center for Drug Design, College of Pharmacy, University of Minnesota, Minneapolis, MN 55455, USA; ssenawee@umn.edu (S.S.); jxie@umn.edu (J.X.); 2Laboratory of Biochemical Pharmacology, Department of Pediatrics, Emory University School of Medicine, Atlanta, GA 30322, USA; haijuan.du@emory.edu (H.D.); huanchun.zhang@emory.edu (H.Z.); karen.kirby@emory.edu (K.A.K.); philip.tedbury@emory.edu (P.R.T.); stefanos.sarafianos@emory.edu (S.G.S.); 3Children’s Healthcare of Atlanta, Atlanta, GA 30322, USA

**Keywords:** Hepatitis B virus, capsid assembly modulators, virtual screening, protein–protein interaction, pharmacophore modelling

## Abstract

Hepatitis B virus (HBV) capsid assembly modulators (CpAMs) have shown promise as potent anti-HBV agents in both preclinical and clinical studies. Herein, we report our efforts in identifying novel CpAM hits via a structure-based virtual screening against a small molecule protein-protein interaction (PPI) library, and pharmacophore-guided compound design and synthesis. Curated compounds were first assessed in a thermal shift assay (TSA), and the TSA hits were further evaluated in an antiviral assay. These efforts led to the discovery of two structurally distinct scaffolds, ZW-1841 and ZW-1847, as novel HBV CpAM hits, both inhibiting HBV in single-digit µM concentrations without cytotoxicity at 100 µM. In ADME assays, both hits displayed extraordinary plasma and microsomal stability. Molecular modeling suggests that these hits bind to the Cp dimer interfaces in a mode well aligned with known CpAMs.

## 1. Introduction

Hepatitis B virus (HBV) chronically infects approximately 260 million people worldwide and causes chronic liver diseases and hepatocellular carcinoma [1]. Two classes of drugs, the direct-acting nucleos(t)ide analogues (NAs) targeting the viral reverse transcriptase (RT), and the immunomodulating interferon alpha (IFN-α), have been approved for the treatment of chronic HBV infection [2]. While NAs are largely effective in reducing viral load, the treatment is not curative due to their inability to eliminate HBV covalently closed circular DNA (cccDNA) which is the reservoir for persistent HBV infection [3]. On the other hand, IFN therapies have only been effective in a minority of patients and exhibit severe side effects, which has hindered their use in clinics [4,5]. Thus, complementary approaches are needed for eradicating or functionally inactivating cccDNA to achieve functional HBV cure [2,6].

The HBV capsid is composed of 120 core protein (Cp) dimers that primarily assemble to form capsids with T = 4 icosahedral symmetry (~95%), with a small number of T = 3 capsids having been observed in HBV-infected human liver samples and in vitro-assembled capsids from recombinant HBV Cp [7,8,9,10]. HBV genome synthesis from pgRNA via reverse transcription is highly dependent on properly assembled viral capsids [11]. In addition, Cp is also implicated in transporting the HBV genome to the nucleus and epigenetic regulation of HBV gene expression [12,13]. Therefore, Cp represents an attractive target in anti-HBV drug discovery [14,15]. Small molecules can destabilize the HBV core protein, increase the capsid assembly rate, and form either aberrant or empty, nonfunctional capsid particles [15]. In recent years, a few major chemical classes have been reported as novel capsid assembly modulators (CpAMs) (Figure 1): heteroaryldihydropyrimidines (HAPs), phenylpropenamides (PPAs), sulfamoylbenzamides (SBAs), sulfamoylpyrroloamides (SPAs), and glyoxamoylpyrroloxamides (GLPs) [15,16]. Among them, HPAs, PPAs, and SBAs are the most studied [17]. 

Structurally, a HBV Cp Y132A “trimer of dimers” in complex with HAP_R01 has been characterized at a resolution of 1.95 Å [18]. HAPs promote Cp mis-assembly by binding to the Cp dimer–dimer interfaces. Three dimer–dimer interfaces accommodate one ligand molecule each and another three molecules bind to similar pockets within the dimers for a total of six molecules in this structure (Figure 2A) [18]. HAPs are known to cause aberrant HBV particles instead of canonical capsids [4,19]. Such structural changes are due to a slightly altered dimer–dimer orientation and multiple hydrophobic contacts between the ligand and the residues in the pocket [20].

Despite the structural differences, all these CpAMs bind to the same pocket at the Cp dimer–dimer interfaces and share common interactions. The phenyl ring A which exists as part of the amide of PPAs, SBAs, SPAs, and GLPs resides in a hydrophobic pocket at the dimer–dimer interface (Figure 2B) [18]. When HAPs are bound, the pocket is occupied by the phenyl moiety at position 4 (ring A) of the 1,4-dihydropyrimidine core. The binding of CpAMs also features a key hydrogen bond with W102, where the dihydropyrimidine N-3 of HAP_R01 and the benzamide oxygen of the other chemotypes act as the hydrogen bond acceptors [18].

In recent years, computational molecular modeling has emerged as a vital part of drug discovery efforts allowing one to perform rapid screening of compound libraries across a wide range of biological targets [21]. In accordance with the advancement of machine learning related to medicinal chemistry [22,23], development of virtual compound libraries is also being carried out by numerous institutions and chemical vendors to ease the process of finding hit compounds in a rapid and cost-effective manner and with an acceptable accuracy. Among the available in silico approaches for hit generation, virtual screening of commercial small molecule compound libraries remains popular [24]. Successful application of such methods relies heavily on the available structural information of the target protein and/or on known bioactive compounds.

During the expansion of the discovery efforts of new chemotypes as potential HBV CpAMs, we turned our attention to rapid identification of novel chemotypes that bind at the same site as HAP_R01 [18]. To this end, structure-based virtual screening was performed using the known protein structure in complex with HAP_R01 ligand (PDB ID: 5WRE). Since CpAMs typically target protein–protein interactions (PPIs) at Cp dimer–dimer interfaces, our screening used a commercial small molecule PPI library. The virtual screening workflow is depicted in Figure 3 and will be discussed in detail under Section 2, Materials and Methods.

To further our aim of identifying new CpAM chemotypes, we carried out hit expansion via a pharmacophore-based approach. Pharmacophore modeling has demonstrated its value in drug design over the last few decades with significant success in hit identification [25]. Such an approach relies heavily on ligand information, rather than the knowledge on the target protein structure. We selected three distinct chemotypes which were previously reported as active HBV CpAMs to generate the pharmacophore hypothesis: ZW-935 (identified by our group, EC_50_ = 0.21 µM) [26], Janssen (disclosed by Janssen Sciences Ireland UC) [27], and GLP-26 [28,29] (Figure 4).

Significantly, a few common structural features among these active ligands allowed us to formulate a tentative pharmacophore model based on their ligand alignments (Figure 4) using the molecular modeling program “Phase” (Schrödinger Inc., New York, NY, USA) [30,31]. These features are denoted as A (acceptor), D (donor), H (hydrophobe) and R (aromatic ring). At this point, instead of conducting the typical virtual screening against a compound library, we chose to pursue a pharmacophore-guided design and synthesis of a small library with numerous complex heterocyclic cores conforming to the common pharmacophore features. Similar approaches have been used by others in medicinal chemistry [32,33,34,35,36,37,38]. Designed compounds are shown in Figure 5. The synthetic tractability from a common intermediate was considered during the process for rapid access to final compounds. Preserved pharmacophore features are shown in the same color as used in the pharmacophore model (Figure 5 vs. Figure 4). The proposed analogs were subjected to molecular docking (PDB ID: 5WRE) using Schrödinger Maestro 2019-1. Compounds were assessed based on both the docking score and the desired binding modes with key residues such as W102 and halogen binding pocket residues. Selected compounds were synthesized and evaluated in our assays. 

Herein, we present the identification of novel HBV CpAM hits using the aforementioned structure and ligand-based approaches. Overall, a total of 106 compounds were curated and assessed using a thermal shift assay (TSA) [39]. TSA hits were further tested in an antiviral assay to confirm potencies. The two best hits, ZW-1841 and ZW-1847, were also assessed for ADME properties, including metabolic stability, plasma stability and aqueous solubility, to further assess the hit quality. Molecular modeling was performed to help understand the binding modes of the confirmed hits.

## 2. Materials and Methods

A small molecule PPI library with 40,640 compounds was selected from Enamine (Enamine PPI library code: PPI-40. The same library is also available through MedChemExpress under the catalog number HY-L0066V). In-house compounds were designed based on the pharmacophore model and synthesized as described in the chemistry section.

### 2.1. Virtual Screening Workflow

#### 2.1.1. Protein Preparation

The virtual screening was carried out using Schrödinger Maestro 2019-1. The crystal structure of Hepatitis B virus core protein Y132A mutant in complex with HAP_R01 (PDB ID: 5WRE) was obtained from the Protein Data Bank as reported by Zhou et al. [18]. There are six HAP_R01 molecules bound to dimer–dimer interfaces in the crystal structures of core protein Y132A hexamer. This model was subjected to Protein Preparation Wizard (Schrödinger Inc.) [40]. During the preprocessing, missing hydrogens were added to the structure and missing side chains and missing loops were added using Prime. Then, the structure was minimized using OPLS 2005 force field to optimize hydrogen bonding network and converge heavy atoms to the RMSD of 0.3 Å. RMSD of 0.3 Å was used during the protein preparation as a measure of the average distance between the atoms (usually the backbone atoms) of superimposed proteins. In this case, RMSD was measured with respect to the 5WRE protein data bank structure. This minimized protein hexamer was further processed to docking purposes by deleting unwanted het molecules, solvent molecules and anions. Finally, amino acid chains A, D, E, and F were deleted along with HAP_R01 molecules in their interfaces. This resulted the prepared protein with chains B and C with a single HAP_R01 molecule residing in the interface.

#### 2.1.2. Receptor Grid Generation

The receptor grid generation tool in Maestro (Schrödinger Inc.) was used to define the active site around the HAP_R01 ligand. Ligand was manually picked during this step, in order to define the active site and to allow the program to exclude the ligand during the grid generation step. Van der Waals scaling factor and partial charge cutoff were kept unchanged from the default values of 1.0 and 0.25 to ensure that only the non-polar atoms are scaled, as scaling the radii of polar atoms has a deteriorating effect on the calculation of accurate hydrogen bonds between the receptor and the ligand. Size of the ligands was set to the similar size to the workspace ligand; HAP_R01 and receptor grid was generated.

#### 2.1.3. Ligand Preparation

Ligand preparation was performed for the commercial and in-house compound library using LigPrep function in Schrödinger Maestro. First, all the ligands were imported to the workspace as a single sdf file. During the process, conformers were generated, possible protonation states were created at pH of 7 ± 2 while retaining specified chiralities as determined by the commercial sdf library. A wide pH range was selected to generate all possible protonated/deprotonated states of the ligands. This was done in order to obtain more virtual hits by generating more conformers of the ligands. The LigPrep output was exported to the workspace in Maestro format and was used for virtual screening.

#### 2.1.4. Virtual Screening

The Schrödinger Maestro Ligand Docking function (Glide) was employed for virtual screening using the prepared receptor grid and prepared ligands [41,42,43]. The Van der Waals radii of nonpolar atoms for each of the ligands were scaled by a factor of 0.8. For the first round, Glide SP (standard precision) was selected to rapidly narrow down the search to a smaller number of ligands with a reasonable docking score. During the docking process, flexible ligand sampling was selected. Options to add Epik state penalties to docking score, to reward intramolecular H bonds, and to enhance planarity of conjugated pi groups were selected during the calculation. The output of this calculation resulted in 4398 structures with a docking score >8.0 (Output group1). This group was resubjected to Glide XP docking to obtain more precise results using the Maestro Ligand Docking function (vide supra), the only difference being the use of XP (extra precision) during the calculation to improve the accuracy. Based on both docking scores and preferred binding modes, a final collection of 100 compounds from the commercial PPI-40 library were selected for biological evaluation. 

#### 2.1.5. Pharmacophore Modelling

The Schrödinger Maestro “develop pharmacophore model” function (Phase) was employed to generate a pharmacophore model using selected active molecules [30,31]. ZW-935, Janssen, and GLP-26 were drawn in the 2D-sketcher in Maestro and the ligands were prepared using the LigPrep function (vide supra). While Phase can (and will as a default) automatically find the best alignment and common features, it is beneficial to manually pre-align ligands. Therefore, the ligands were processed through the Flexible Ligand Alignment tool before importing into Phase. Then, the aligned ligands were subjected to pharmacophore modeling. Given that all three ligands are known potent CpAMs, they were defined as “actives” during the process. Multiple hypotheses were generated upon completion of the process. The best hypothesis was chosen based on the Survival score which is the weighted combination of the vector, site, volume scores, and a term for the number of matches. The hypothesis was further inspected visually for overlaps of pharmacophore properties between the three selected ligands. After identifying overlapping pharmacophore features through pharmacophore modeling, 6 chemotypes with complex heteroaryl core structures were synthesized by preserving the common features and are described below.

### 2.2. Chemistry

All analogs were synthesized using procedures described in Scheme 1, fully characterized with ^1^H NMR, ^13^C NMR, and HRMS, and displayed a purity of ≥95% as determined by HPLC. Detailed synthetic procedures and compound characterization data are included in Appendix A.

During the synthetic planning, we recognized compounds **3** and **4** (Scheme 1) as versatile intermediates with a potential to be rapidly converted into numerous complex heterocyclic cores. The intermediate **3** was conveniently accessed via an aqueous phase reaction between 4-fluoro-3-methylaniline (**1**) and 2,2,6-trimethyl-4H-1,3-dioxin-4-one (**2**) [44]. Base-mediated addition of 2-cyanoacetamide to the intermediate **4** followed by a thermal cyclization yielded the 6-methyl pyridine derivative ZW-1840 [45]. The intermediate **4** was further diversified to rapidly generate fused heteroaromatic cores by reacting it with the corresponding amines (**5**, **6** and **7**) under thermal conditions [46] to yield ZW-1841, ZW-1842, and ZW-1843. Pyrazolopyridone chemotype ZW-1847 was obtained via a three-step process starting from the intermediate **3** [47]. First, compound **3** was added to CS_2_ in base-mediated fashion followed by a methylation to yield oxoketene dithioacetal intermediate **8**. The cyclocondensation reaction of 2-(bis(methylthio)methylene)-3-oxo-N-arylbutanamide (**8**) with cyanoacetamide using NaO^i^Pr as base under reflux condition afforded functionalized pyridone (ZW-1845). Further, [3 + 2] cyclocondensation reaction of the pyridone with hydrazine in the presence of alcohol yielded the final pyrazolopyridones compound ZW-1847.

### 2.3. Biology

#### 2.3.1. Reagents 

Biologicals. HuH-7 (JCRB Cell Bank) cells were maintained in complete media [Dulbecco’s Modified Eagle Medium (DMEM), 10% fetal bovine serum (FBS) and incubated at 37 °C with 5% CO_2_.

#### 2.3.2. Cp Purification

A gBlock Gene Fragment coding for the 149 amino acid assembly domain of HBV capsid protein with an added C-terminal cysteine (C150) [20,48,49] with NdeI and BamHI restriction sites was synthesized by Integrated DNA Technologies and cloned into the pET11a expression vector (Novagen). HBV C150 was expressed and purified as previously described [48,49,50], with minor modifications. The C150 expression plasmid was transformed into BL21 (DE3) *E. coli*, grown at 37 °C to an OD600 of ~0.8, and induced for 3 h with 1 mM IPTG at 37 °C. Cells were pelleted and resuspended in 50 mM Tris (pH 7.5), 1 mM EDTA, 20 mM 2-mercaptoethanol (2-ME), 1 mM PMSF, 150 µg/mL lysozyme, and 0.2 mg/mL DNase I. The suspension was incubated on ice for 30 min and lysed by sonication. Polyethylenimine (PEI) was added to a final concentration of 0.15% *w*/*v* to precipitate DNA, and the lysate was centrifuged at 16,000× *g* for 1 h. Ammonium sulfate was added to the supernatant to 40% saturation. The solution was gently stirred for 1 h, then centrifuged at 16,000× *g* for 1 h. The pellet was resuspended in Buffer A [100 mM Tris (pH 7.5), 100 mM NaCl, 10 mM 2-ME] to ~10 mg/mL, centrifuged at 16,000× *g* for 20 min, loaded onto a Buffer A-equilibrated HiLoad 26/60 Superdex 200 prep grade (GE Healthcare) column, and eluted at 2.5 mL/min. Fractions were pooled based on the chromatogram and SDS-PAGE, concentrated to ~5 mg/mL, and dialyzed into Buffer N [50 mM sodium bicarbonate (pH 9.6), 10 mM 2-ME]. Solid urea was added to 3 M and stirred for 1 h at 4 °C. The solution was loaded onto a Buffer N-equilibrated HiLoad 26/60 Superdex 200 prep grade column and eluted at 2.5 mL/min. Fractions containing the C150 dimer (C150_2_) were pooled, concentrated, and stored at −80 °C. Final protein concentration was determined spectrophotometrically using an extinction coefficient of 60,900 [49].

#### 2.3.3. Thermal Shift Assay

The thermal shift assay was performed as previously described [39]. Briefly, in a final reaction volume of 20 µL, 10 µL of C150_2_ (15 µM) in Buffer N were mixed with 10 µL of assembly buffer [100 mM HEPES (pH 7.5), 1 M NaCl] containing 2×Sypro Orange Protein Gel Stain (Life Technologies, Carlsbad, CA, USA). Compounds were added at a final concentration of 20 µM, and reactions contained 1% DMSO. Samples were heated in a QuantStudio 3 Real-Time PCR system (Thermo Fisher Scientific, Waltham, MA, USA) from 25 to 95 °C with a heating rate of 0.2 °C/10 sec. Melting curves were analyzed with Protein Thermal Shift™ Software version 1.3 (Thermo Fisher Scientific).

#### 2.3.4. Antiviral Assays

HuH-7 cells were seeded in 96-well plates (2.5 × 10^4^ cells/well). The following day, the cells were transfected with an HBV replication luciferase reporter plasmid (details in manuscript in preparation). A master mix of transfection reagent was produced sufficiently for all wells to be transfected. This was then diluted in 100 µL/well DMEM; the medium was removed from the plate and 100 µL/well diluted transfection mix added. Compounds were added at the time of transfection, then after 24 h, the transfection mix was removed, and fresh medium containing compounds was added. After 5 additional days, the medium was harvested, and the luciferase activity was measured using the Nano-Glo Luciferase Assay System (Promega, Madison, WI, USA). Values were plotted in GraphPad Prism 5 and analyzed with the log (inhibitor) vs. normalized response–variable slope equation.

#### 2.3.5. Cytotoxicity Assays

HepG2 cells were plated in complete media and treated with compounds as in the antiviral screening above. At the end of treatment duration, cell viability was assessed with the Cell Proliferation Kit II (XTT) (Roche) according to the manufacturer’s instructions. Values were plotted in GraphPad Prism 5 and analyzed with the log (inhibitor) vs. normalized response–variable slope equation.

#### 2.3.6. ADME Assays

*Aqueous solubility*. The aqueous solubility of each selected compound was determined in Dulbecco’s Phosphate-Buffered Saline (DPBS) under thermodynamic solubility conditions. Briefly, a saturated solution was made by adding DPBS to the solid compound. The mixture was shaken at 200 rpm for 72 h at ambient temperature to allow equilibrium between the solid and dissolved compound. The suspension was then filtered through a 0.45 µM PVDF syringe filter and the filtrate was collected for immediate analysis using LC/MS/MS.

*Plasma stability.* The plasma stability assay was performed in triplicate by incubating each selected compound (1 µM final concentration) in normal mouse (CD-1) and human plasma diluted to 80% with 0.1 M potassium phosphate buffer (pH 7.4) at 37 °C. At 0, 0.5, 1, 2, and 3 h, aliquots of the plasma mixture were taken and quenched with 3 volumes of acetonitrile containing 0.1% formic acid. The samples were then vortexed and centrifuged at 14,000 rpm for 5 min. The supernatants were collected and analyzed by LC/MS/MS to determine the remaining percentage or half-life time (t_1/2_).

*Microsomal stability.* The in vitro microsomal stability assay was conducted in triplicate in mouse and human liver microsomal systems, which were supplemented with nicotinamide adenine dinucleotide phosphate (NADPH) as a cofactor. Briefly, a compound (1 µM final concentration) was spiked into the reaction mixture containing liver microsomal protein (0.5 mg/mL final concentration) and MgCl_2_ (1 mM final concentration) in 0.1 M potassium phosphate buffer (pH 7.4). The reaction was initiated by addition of 1 mM NADPH, followed by incubation at 37 °C. A negative control was performed in parallel without NADPH to reveal any chemical instability or non-NADPH dependent enzymatic degradation for each compound. At various time points (0, 5, 15, 30 and 60 min), 1 volume of reaction aliquot was taken and quenched with 3 volumes of acetonitrile containing 0.1% formic acid. The samples were then vortexed and centrifuged at 15,000 rpm for 5 min at 4 °C. The supernatants were collected and analyzed by LC/MS/MS to determine the remaining percentage and in vitro metabolic half-life (t_1/2_).

## 3. Results

All 106 compounds were first evaluated in the TSA to screen for compounds which have an effect on HBV capsid assembly. Although not quantitative, the assay provides an effective measure of the impact of compounds on HBV capsid assembly, as it is performed under assembly conditions (in the presence of high salt concentration) [39]. Compounds were screened at 20 µM final concentration, and melting curves were analyzed for melting temperature (T_m_) shifts or any changes in the ratio of peak 1 (left-most peak) to peak 2 (right-most peak). We chose to inspect the curves visually for this screen due to the complexities of the melting curves. From this assay, 11 hits with a significant impact on HBV capsid assembly were selected (representative graphs for selected hit compounds are shown in Figure 6 and Appendix A). Examples of TSA melting curves for compounds that did not have an effect on HBV capsid assembly are shown in Appendix A. We further tested these hits in an antiviral assay which measures HBV replication to determine EC_50_ in cell culture. These compounds were also evaluated in a cytotoxicity assay. The antiviral and cytotoxicity data for the selected 11 compounds are presented in Table 1.

In addition, we also characterized ZW-1841 and ZW-1847 in three major in vitro ADME (absorption, distribution, metabolism and excretion) assays: aqueous solubility, plasma (human and mouse) stability and microsomal (human and mouse) stability. The results are shown in Table 2.

## 4. Discussion

Although the screening set (106 compounds) was curated using two distinct approaches, compounds with significant antiviral activity were identified from both groups (Table 1): ZW-1840, ZW-1841, ZW-1845 and ZW-1847 from pharmacophore-guided synthesis; and ZW-1888 and ZW-1929 from virtual screening. Overall, the pharmacophore-based approach generated two hits with antiviral potencies in single-digit µM concentrations: ZW-1841(EC_50_ = 6.6 µM) and ZW-1847 (EC_50_ = 3.7 µM), whereas the most potent hit from virtual screening, ZW-1888, inhibited HBV with an EC_50_ of 17.2 µM (Table 1). In the cytotoxicity assay, ZW-1888 was moderately cytotoxic (CC_50_ = 47 µM), while both ZW-1841 and ZW-1847 were nontoxic at concentrations up to 100 µM. These results clearly define ZW-1841 and ZW-1847 as our best hits and prompted us to assess their major ADME properties. As shown in Table 2, both compounds demonstrated excellent plasma and microsomal stability, properties well aligned with favorable bioavailability. However, the aqueous solubility for both hits needs improvement (Table 2), which will be among the main goals of future hit optimization efforts via analog synthesis.

To better understand the observed potencies, we analyzed binding modes of the three active compounds. We used the best output poses obtained from either the virtual screening or pharmacophore guided design (vide supra) and analyzed them using PyMOL visualization software. The best output pose was determined based on the docking score of the individual output poses as well as the possible bad interactions such as steric clashes between the ligand and the protein. We first looked at the individual binding mode of ZW-1841, ZW-1847, and ZW-1888, which consist of benzimidazolopyrimidine, pyrazolopyridone, and 2*H*-benzoxazin-3(4*H*)-one core structures along with a benzamide moiety. The predicted binding modes for the above compounds are shown in Figure 7A–C.

The molecular models predict that all the three inhibitors with EC_50_ < 20 µM have a halogenated phenyl amide located in the halogen binding pocket (Figure 7). For ZW-1841, the benzamide carbonyl and the amide N–H are predicted to form a hydrogen bond with W102 (from chain A) and the oxygen atom of T128 (from chain B), respectively. Interestingly, the fused heteroaromatic core of the modeled ZW-1841 is flipped out from the hydrophobic sub-pocket to the solvent-exposed region (Figure 7A). This is likely due to potential steric clashes between three fused-ring moiety and the amino acid residues on the hydrophobic pocket. By contrast, the pyrazolopyridone core of ZW-1847 is positioned to occupy the hydrophobic sub-pocket and reach the proximity of F23 and Y118 (Figure 7B). The conserved H-bond with T128 is formed by the benzamide N–H. The superior activity of ZW-1847 compared to the other two hits may arise from the additional H-bonding interaction between N(5)–H of the pyridone core and the oxygen atom of Y118, as shown in Figure 7B. The aromatic residues Y132, F23, Y118, and F122 at the Cp dimer–dimer interface are crucial for icosahedral capsid assembly [7,18,51]. Selective targeting of those residues, particularly Y118, has proven successful in developing selective inhibitors that disrupt HBV capsid assembly [52].

The predicted binding mode of ZW-1888 is also largely consistent with the binding modes of reported CpAMs. In this binding mode, the hydrophobic halogen binding pocket accommodates the difluoromethyl phenyl group, while the fused heterocyclic core extends to the second hydrophobic sub-pocket (Figure 7C). Although the compound maintains the H-bonding interaction with W102 via its amide carbonyl, the other typically conserved H-bonding with T128 is absent, presumably because the amide N–H instead forms an intramolecular H-bonding with the ring oxygen of the 2H-benzoxazin-3(4H)-one core structure (depicted as a red dotted line). The loss of the H-bond with T128 may account for the reduced potency of ZW-1888 when compared with the other two hits.

Finally, Figure 7D shows the overlay of the best docking pose of our most potent hit, ZW-1847, and that of a known CpAM, ZW-935 [26]. The overlay revealed that the two compounds are exceptionally well aligned at both the carboxyamide moieties and heteroaromatic cores. In both cases, halogenated phenyl amide extends to the halogen binding pocket and maintains the major interactions with amino acid residues such as W102 (other interactions are omitted for clarity), while keeping their heteroaryl cores located in the second hydrophobic sub-pocket.

## 5. Conclusions

To identify novel HBV CpAM hits, we conducted a structure-based virtual screening against a commercial small molecule PPI library, and a pharmacophore-guided synthesis of a few novel chemotypes. In the end, although active compounds were identified via both approaches, the pharmacophore approach produced two scaffolds as novel HBV CpAMs. These two hits, ZW-1841 and ZW-1847, displayed EC_50_ values of 6.6 µM and 3.7 µM, respectively, in antiviral assays, and showed no toxicity at concentrations up to 100 µM in cytotoxicity assays. Molecular modeling revealed that the predicted binding modes of these compounds are well aligned with that of the known CpAMs. In assays measuring major ADME properties, both hits demonstrated excellent plasma and microsomal stability.

## Data Availability

The data presented in this study are available in the Appendix A.

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
