# Peer review of "Discovery of New Small Molecule Hits as Hepatitis B Virus Capsid Assembly Modulators: Structure and Pharmacophore-Based Approaches"

_viruses, 2021, doi:10.3390/v13050770_

Round 1

Reviewer 1 Report

This report shows the design of anti-HBV chemicals to inhibit HBV capsid formation. I would say that chemistry is very fine but virological assessment is extremely poor.

The author must show inhibition activity of the compounds in terms of capsid formation and viral replication.

Current context is not good enough for publication in the journal “Viruses”.

Reviewer 2 Report

The hepatitis B virus is an important cause of acute and chronic liver disease and yet with suboptimal treatment. Studies aims to identify potential antiviral drugs are hence of great interest and importance. Current study is one of the kinds that use state of approaches to screen or design compounds interfere with HBV capsid formation. With the selected hits, the author has also tried to provide some basic pharmacological and biological characterization, which added great value to the screening part of the study. However, there remain several issues need to be addressed.

Major:

  1. Line 42. The author need to revise the text and use more recent reference. HBV capsids with either T=3 or T=4 icosahedral symmetry were identified from patient /transgenic mouse plasma and spontaneous assembled capsid expressed from bacteria.
  2. It is not clear at all what antiviral assay was performed. The author should provide information about the model and makers used. A HBV replication model with luciferase as readout raise the question if the antiviral effect is related to the original screening goal, as capsid inhibitor typically do not affect virus protein expression.  

Author Response

Line 42. The author need to revise the text and use more recent reference. HBV capsids with either T=3 or T=4 icosahedral symmetry were identified from patient /transgenic mouse plasma and spontaneous assembled capsid expressed from bacteria.

Response: We thank the reviewer for this excellent suggestion and have updated the text to state that HBV Cp assembly can result in capsids with both T=3 and T=4 icosahedral symmetry. We also include updated references.

It is not clear at all what antiviral assay was performed. The author should provide information about the model and makers used. A HBV replication model with luciferase as readout raise the question if the antiviral effect is related to the original screening goal, as capsid inhibitor typically do not affect virus protein expression.  

Response:

This is addressed in our response to the comment raised by reviewer 1.

Reviewer 3 Report

The manuscript by Senaweera et al. reports on the discovery of new small molecule hits that target HBV capsid assembly. HBV Cp assembly modulators represent an orthogonal class of potential drugs that show potent anti-viral activity in pre-clinical and clinical settings. Overall, the manuscript is well-written. It reports the results of a combination of computational and experimental screening for drugs that affect HBV Cp assembly. Two ideal candidates were identified (reported in Table 2). While not ideal yet, these molecules can serve as a building block for further development/ pharmacologic optimization. Broadly, this reviewer is satisfied with the results reported in the manuscript, but would like to provide following suggestions to improve the overall quality and cross-disciplinary appeal of the manuscript:

  1. In thermal shift assay, the authors should discuss background fluorescence of individual drugs. Was the background fluorescence measured and corrected for? If so, how was that done? It is known that for many fluorophores, fluorescence is highly temperature-dependent.
  2. The 3-step thermal denaturation profile indicate that there might be 2-processes operative. It also appears that different compounds have different denaturation profiles (compare Fig. 6B vs. 6A&C). Assuming these differences are not coming from the drugs themselves, it is likely that different drugs affect different steps (maybe oligomerization step vs. monomer unfolding).
    1. The authors may wish to discuss this aspect.
    2. TSA profiles for all 106 compounds should be provided in the SI. This will allow the readers to get a better clarity with regards to individual drugs’ mode of action.
  3. For the aqueous solubility assay, the mixture was shaken for 72 hours, then filtered through PVDF syringe and filtrate was analyzed with LC/MS/MS. Was the LC/MS/MS step done immediately after the filtration or there were some waiting periods (like 24 hours after filtration)? Mentioning this would be helpful to better grasp solubility measurements.

Author Response

In thermal shift assay, the authors should discuss background fluorescence of individual drugs. Was the background fluorescence measured and corrected for? If so, how was that done? It is known that for many fluorophores, fluorescence is highly temperature-dependent.

Response: We thank the reviewer for the excellent point. In our experience with multiple proteins and various chemical libraries, we have had instances with similar problems described by the reviewer: for example, in another study we encountered a small number of compounds that fluoresce to a variable extent in the absence of protein. This was indeed a confounding factor that made it difficult to determine whether there is binding between compounds and the target. In such cases we typically analyze the compounds by determining the antiviral activity and if biophysical characterization is required we try methods such as microscale thermophoresis. In the case of the compounds listed in this study we did not observe any significant interference by compound fluorescence.

We also agree with the reviewer that fluorescence is temperature dependent; for this reason we carefully control the temperature of the experiment and ensure consistency between the extensive number of replicates (multiple technical and biological replicates).

The 3-step thermal denaturation profile indicate that there might be 2-processes operative. It also appears that different compounds have different denaturation profiles (compare Fig. 6B vs. 6A&C). Assuming these differences are not coming from the drugs themselves, it is likely that different drugs affect different steps (maybe oligomerization step vs. monomer unfolding).

  1. The authors may wish to discuss this aspect.
  2. TSA profiles for all 106 compounds should be provided in the SI. This will allow the readers to get a better clarity with regards to individual drugs’ mode of action.

Response: These are excellent points. Indeed, there can be differences in the thermal denaturation profiles of various Cp-compound complexes. We are generally cautious in trying to assign mechanistic information on the basis of the shape of denaturation curves, but in our experience there can be valuable clues that can be used. Generally speaking, the most interesting change and less prone to misinterpretation is whether a compound “stabilizes” or “destabilized” the target. We have worked extensively on identifying potential correlations between the magnitude of thermal shift and effect on biological activities in another HIV-related system.

However, for the purposes of this study, thermal shift assays were used to flag compounds to be tested for their effect on the biological activity. Our lab uses extensive mechanistic tools to determine the precise effect of antivirals on the mechanism and structure of the targeted compound, which are beyond the scope of this study that aims to identify hit compounds for further lead development.

Given the complexity of the various curves that may provide uninterpretable information as well as logistics constraints we only provide a sample of TSA curves for hit compounds that bind to HBV Cp and affect capsid formation and stability in Figure 6 and Supplementary Figure 1A-E. We also provide 5 examples of TSA curves for compounds that were screened but did not bind to HBV Cp (DMSO and compounds curves are highly similar with few differences) in Supplementary Figure 1F-J. We have updated the text accordingly.

For the aqueous solubility assay, the mixture was shaken for 72 hours, then filtered through PVDF syringe and filtrate was analyzed with LC/MS/MS. Was the LC/MS/MS step done immediately after the filtration or there were some waiting periods (like 24 hours after filtration)? Mentioning this would be helpful to better grasp solubility measurements.

Response: Yes, the filtered samples were immediately analyzed using LC/MS/MS. We have made the change accordingly to reflect the timing of LC/MS/MS analysis of the samples.

Reviewer 4 Report

The manuscript by Senaweera et al. seeks to identify hits for potential inhibitor compounds against the assembly of the hepatitis B virus capsid. The authors’ approach employs structure-based virtual screening and pharmacophore-guided design to identify a tractable number of compounds for synthesis and testing. Both screening and design produced active compounds, and ultimately two of these were reported as novel hits for HBV capsid protein assembly modulators (CpAM), showing appreciable antiviral potency without cytotoxic effects.

The manuscript is clear and detailed, and does a good job of explaining the protocols for virtual screening and pharmacophore-guided design of compounds for those unfamiliar with the approaches. That predicted compounds were shown to be effective in cells is compelling and validates the computational methodology.  The results, suggesting two new CpAM hits ready for optimization, are important, as the search for anti-capsid compounds is a very active area in the hepatitis B field, with new therapeutic options badly needed by patients. I recommend the manuscript for publication following minor revisions. 

The authors use the 5WRE protein data bank structure for screening, which is a trimer of dimers bound to HAP_R01. The dimers contain a mutation Y132A that prevents capsid assembly, yet in the protein data bank structure, the dimers are apparently sufficiently “assembled” to reproduce the CpAM binding pocket. How does the pocket in 5WRE compare with the pocket in the capsid, either apo or bound to HAP? Both such capsid structures are available for comparison. The authors note that HAPs induce changes in dimer-dimer orientations (lines 63-64), and the trimer of dimers must be a flat extended lattice in its crystal versus the capsid which has trimers of dimers incorporated into a curved surface. Probably 5WRE was selected due to resolution, but a comment on why this structure was selected rather than one that is representative of a capsid protein oligomer known to exist physiologically is desirable. The authors note that virtual screening is dependent on the protein structure (lines 89-91), so would the trimer of dimers and capsid structure produce equivalent results from screening? 

When the authors return to their screening models to rationalize binding modes, they say they used the “best” output poses for visualization (lines 448-449). What does “best” mean in this context, and were there other poses that scored sufficiently high by this metric that were also worth visualizing? If so, do they suggest alternative binding modes that should be considered? The authors note that the “best” poses produced binding modes that were largely consistent with that of reported CpAMs (lines 439-440); is this outcome because a CpAM-bound crystal was used for the screening? Surely the choice of structure can lead to bias in such results. In general, there are limitations to these methods, and the authors should point them out where appropriate. 

In the methods section, the authors provide some numerical values important for reproducibility of the virtual screening process. In line 162, they mention an RMSD value, but do not clearly indicate their reference. RMSD is a measure of distance between atoms, so what is RMSD being measured with respect to? In line 172, they mention a partial charge cutoff of 0.25. Is this a distance-based cutoff? What is the unit? Depending on the unit, this value appears extremely short for a distance-based truncation of electrostatic interactions, which are surely critical to reproducing key features of the binding mode like hydrogen bonds. In line 180, they mention assigning protonation states to the protein at pH 7+/- 2. Does this mean that structures representing the range of protonation states possible in this large pH range of 5 to 9 were examined? Surely pH 5 and pH 9 give rise to very different results, changing the locations of hydrogens and charged residues within the binding site. Are these protonation states relevant biologically?

Author Response

The authors use the 5WRE protein data bank structure for screening, which is a trimer of dimers bound to HAP_R01. The dimers contain a mutation Y132A that prevents capsid assembly, yet in the protein data bank structure, the dimers are apparently sufficiently “assembled” to reproduce the CpAM binding pocket. How does the pocket in 5WRE compare with the pocket in the capsid, either apo or bound to HAP? Both such capsid structures are available for comparison. The authors note that HAPs induce changes in dimer-dimer orientations (lines 63-64), and the trimer of dimers must be a flat extended lattice in its crystal versus the capsid which has trimers of dimers incorporated into a curved surface. Probably 5WRE was selected due to resolution, but a comment on why this structure was selected rather than one that is representative of a capsid protein oligomer known to exist physiologically is desirable. The authors note that virtual screening is dependent on the protein structure (lines 89-91), so would the trimer of dimers and capsid structure produce equivalent results from screening? 

Response: As the reviewer has noted Y132A mutation is required to obtain high resolution crystal structures of Cp trimer of dimers by preventing capsid assembly. These crystal structures have facilitated the design of next-generation HAP compounds which would not have been possible with the available low-resolution structures of the fully assembled capsid. As we have described in line 56-63, HAPs promote Cp mis-assembly by binding to the Cp dimer-dimer interfaces, which are present in the Cp trimer of dimers lattice. Therefore, it is justifiable and preferable to perform docking using the dimeric structure rather than the fully assembled capsid.

When the authors return to their screening models to rationalize binding modes, they say they used the “best” output poses for visualization (lines 448-449). What does “best” mean in this context, and were there other poses that scored sufficiently high by this metric that were also worth visualizing? If so, do they suggest alternative binding modes that should be considered? The authors note that the “best” poses produced binding modes that were largely consistent with that of reported CpAMs (lines 439-440); is this outcome because a CpAM-bound crystal was used for the screening? Surely the choice of structure can lead to bias in such results. In general, there are limitations to these methods, and the authors should point them out where appropriate. 

Response:

  • The best output pose was determined based on the docking score of the individual output poses as well as the possible bad interactions such as steric clashes between the ligand and the protein. We have observed some alternative binding modes with significantly lower docking scores and bad interactions. We have updated the text by including the above details.
  • We compared the binding poses of our compounds with the previously known compounds. This was done by visual observation of the interactions between different parts of the molecules and amino acid residues of the protein. The advantage of using a well-resolved liganded structure for virtual screening is that a grid can be defined with high confidence. So yes, the ligand knowledge enables a "biased" approach which is preferred over the completely unbiased random screening, though there is no “bias” in chemotypes--the molecules we docked are structurally diverse and very different from the HAP molecule, the co-crystalized ligand in PDB 5WRE.

    In the methods section, the authors provide some numerical values important for reproducibility of the virtual screening process. In line 162, they mention an RMSD value, but do not clearly indicate their reference. RMSD is a measure of distance between atoms, so what is RMSD being measured with respect to? In line 172, they mention a partial charge cutoff of 0.25. Is this a distance-based cutoff? What is the unit? Depending on the unit, this value appears extremely short for a distance-based truncation of electrostatic interactions, which are surely critical to reproducing key features of the binding mode like hydrogen bonds. In line 180, they mention assigning protonation states to the protein at pH 7+/- 2. Does this mean that structures representing the range of protonation states possible in this large pH range of 5 to 9 were examined? Surely pH 5 and pH 9 give rise to very different results, changing the locations of hydrogens and charged residues within the binding site. Are these protonation states relevant biologically?

    Response:

    • RMSD of 0.3 Å was used during the protein preparation as a measure of the average distance between the atoms (usually the backbone atoms) of superimposed proteins. In this case, RMSD was measured with respect to the 5WRE protein data bank structure. The text was updated accordingly.
    • A partial charge cutoff of 0.25 was used in the receptor grid generation. This is to ensure that only the non-polar atoms are scaled, as scaling the radii of polar atoms has a deteriorating effect on the calculation of accurate hydrogen bonds between the receptor and the ligand. The text was updated accordingly.
    • A pH range of pH 7+/- 2 was used during the ligand preparation step- not during the protein preparation. As the reviewer has stated the operation generates all possible protonated/deprotonated states of the ligands. This was done in order to obtain more virtual hits by generating more conformers of the ligands. The text was updated accordingly.

Round 2

Reviewer 1 Report

As mentioned previously, there is no biological/virological assay in this manuscript, which is not suitable for this kind of virology journal.

Reviewer 2 Report

The revised paper brings new idea the drug development for hepatitis B treatment.